# The quest for a generic bird target to detect the presence of bird in food products and considerations for paleoprotein analysis

**Anne J. Kleinnijenhuis**\* , **Frédérique L. van Holthoon**

Triskelion, Utrecht, The Netherlands

\* anne.kleinnijenhuis@triskelion.nl

**Data Availability Statement:** All relevant data are within the paper and its Supporting information files.

## Abstract

It can be important for consumers to know whether food products contain animal material and, if so, of which species. Food products with animal material as an ingredient often contain collagen type 1. LC-MS/MS (Liquid Chromatography–tandem Mass Spectrometry) was applied as technique to generically detect bird. Unlike for example fish, that have experienced longer divergence times, it is still possible to find generic LC-MS targets for avian type 1 collagen. After theoretical target selection using 83 collagen 1α2 bird sequences of 33 orders and construction of a common ancestor sequence of birds, experimental evidence was provided by analyzing extracts from 10 extant bird species. Two suitable options have been identified. The combination of VGPIGPAGNR and VGPIGAAGNR (pheasant only) covers all investigated birds and was not found in other species. The peptide EGPVGF**p**-GADGR covers all investigated birds, but also occurs in several species of crocodiles and turtles. The presence of the generic peptide (combination) was confirmed in food products, proving the principle, and can therefore be used to detect the presence of bird. Furthermore, it is shown how the use of constructed ancestor sequences could benefit the field of paleo-proteomics, in the interpretation of collagen MS/MS spectra of ancient species. Our theoretical analysis and assessment of reported *Brachylophosaurus canadensis* collagen 1α2 MS/MS data provided support for several previous peptide sequence assignments, but we also propose that our constructed ancestral bird sequence GP**p**GESGAVGPAGPIGSR may fit the MS/MS data better than the original assignment GLPGESGAVGPAGP**p**GSR.

## Introduction

Collagen type 1 forms the structural and mechanical scaffolding of skin, bones, tendons, blood vessel walls, cornea and other connective tissues. It is the most abundant protein in vertebrates [1] and consists of triple helices [2]. Usually the helices are heterotrimers of two collagen 1α1 chains and one collagen 1α2 chain [3], although skin type 1 collagens from several bony fish additionally contain a 1α3 chain in the heterotrimer along with 1α1 and 1α2 [4–7]. Collagen analysis is performed in several fields, e.g. in medical research [8,9], food chemistry [10] and paleoprotein analysis [11–14]. Food products with animal material as an ingredient often

**Funding:** The authors received no specific funding for this work.

**Competing interests:** The authors have declared that no competing interests exist.

contain collagen type 1, either due to its ubiquity and high abundance in (extracts from) animal tissues and/or by the intended addition of gelatin, which is partly hydrolyzed collagen often obtained from skin or bone [15,16]. For religious or lifestyle reasons it may be important for consumers to know whether food products contain animal material and, if so, of which species. To determine animal species, suitable and reliable analytical techniques should be applied, targeting informational biomolecules such as DNA or protein [17]. In order to find generic targets to detect the presence of bird in food products, the main goal of this study, it is necessary to perform a detailed analysis of their molecular evolution in birds and to construct ancestral sequences. An ancestor sequence of avian collagen represents collagen from the past and, compared to sequences of extant birds, will bear a greater resemblance to collagen extracted from fossilized bones of ancient birds and non-avian dinosaur species. Therefore, the approach presented in this study is also relevant to paleontologists investigating the sequence of collagen in fossilized tissues.

The treatment conditions in the gelatin production process destroy most of the DNA, severely reducing the sensitivity of DNA-based methods or leading to the occurrence of false negative results [10]. The performance of ligand binding assays (LBA) depends on the 3D structure of analyte proteins [18], which is affected by gelatin production conditions or by food processing in general. Responsible use of LBA in (processed) food analysis may require knowledge of reagent-analyte interactions at the molecular level to assess the suitability of an assay, especially with regard to its selectivity and the potential change in affinity for analyte proteins that have a changed 3D structure after food processing. It is preferable to use bottom-up protein LC-MS/MS (Liquid Chromatography–tandem Mass Spectrometry) as a detection technique over DNA-techniques and LBA, because the primary protein structures often remain largely intact during food processing. This is beneficial for reliable detection, especially when smaller target peptides are cleaved from the primary structure. Moreover, the largely intact primary structures of collagen contribute to the intrinsic properties of gelatin, i.e. the enabling of gelatinization. Targets that differ in a single amino acid can easily be distinguished using LC-MS/MS by their retention time, the m/z of precursor ions and/or the m/z of product ions after fragmentation [19]. Several strategies can be applied to detect animal species: one strategy is to select unique targets per species, which are absent in other species; another approach is to find generic targets for a whole group of animal species [20], such as birds. The latter approach was followed in this study. The goal of the study was to identify generic bird targets to add these as modules to the TrustGel™ method [21].

Phylogenetic studies indicate an early emergence of the three fibrillar collagen clades A to C, before the eumetazoan radiation [22]. Therefore collagen type 1 chains, belonging to the A clade, are present in all bird species and provide a good basis for finding a generic bird target. Eumetazoan radiation occurred approximately 530 million years ago [23], well before the evolution of modern birds. Approximately 165–150 million years ago, birds evolved from theropod dinosaurs and continuously served as a vehicle for protein evolution. The mass extinction event at the end of the Cretaceous, 66 million years ago, decimated the number of bird species and lineages. After that, birds explosively diversified in more than 10,000 species today [24–26]. The closest extant relatives to birds are, in descending order, crocodiles, turtles and lizards (including snakes and geckos) [27].

Previously, we developed method modules for the individual detection of quantitative porcine and bovine targets [21] and for several fish species [7], amongst others. Generic bird targets should cover as many bird species as possible, but should be different from other animal species. An advantage of a generic target is that fewer transitions need to be added to a targeted, quantitative LC-MS method compared to when species are detected individually, leaving more room for detection of other targets in the same method. A disadvantage is that it

cannot be known whether a generic target truly covers an entire group when genetic information is not available for every species in the group and for a sufficient number of individuals, to cover the variation within a species. However, these aspects will also impair the detection of individual species with unique targets. The theoretical target selection was performed using 83 bird sequences, including the bird species most important in food products, several sequences from related animal groups and database searches. Experimental support was provided by analyzing 10 bird species using non-targeted LC-MS/MS. Targets were selected from collagen 1α2, due to 1) the high abundance of collagen type 1 in general and because 2) less (reliable) genetic information could be retrieved from databases for avian collagen 1α1 than for collagen 1α2. Using collagen 1α1 as target source would reduce the quality of the theoretical target selection process, compared to collagen 1α2. Additionally, gene loss has not been reported for collagen 1α2 in birds, in contrast to collagen 1α1 [28]. Ultimately, one generic target and one target combination met the selection criteria. Their presence was experimentally confirmed in extracts from 10 extant bird species, in chicken soup and in chicken broth, as proof of principle for food products. Finally, it was shown how our combined food chemistry and molecular evolution approach could benefit paleoprotein analysis, in the interpretation of collagen MS/MS spectra from ancient species.

## Materials and methods

Bird products (ostrich, goose, duck, turkey, chicken, pheasant, guinea fowl, pigeon, partridge, quail, various cuts) were purchased from local supermarkets. Collagen from these birds was extracted by placing several grams per product in milliQ water in an oven at 100˚C for 2 days. After centrifugation, 4 ml extract was added to 3 ml pentane, shaken and centrifuged. After 1 hour at < -70˚C the pentane layer was removed. Aliquots were digested with trypsin without reduction or alkylation because the collagen GXY domain, from which the peptide targets of interest were cleaved, contains no disulfide bridges. Chicken soup was homemade and was sampled by taking an aliquot of the gelatinous part, after the soup had cooled. Chicken broth (brand A (1.3% chicken meat powder) and brand B (3.1% chicken meat powder)) and beef broth (2.3% beef extract) tablets were purchased from local supermarkets. Tablets were dissolved in 100 mM ammonium bicarbonate at 95˚C for 3 hours. After centrifugation, aliquots were digested with trypsin. The beef broth sample served as a negative control.

Samples were analyzed using a combination of a UHPLC (Ultimate 3000, Dionex) and a Q-Exactive mass spectrometer (ThermoElectron). Separation was achieved on an Acquity HSS T3 column (2.1 × 100 mm, 1.8 μm, Waters, Milford, PA, USA) at a temperature of 40˚C with an injection volume of 10 μl. Mobile phases consisted of milliQ water (A) and acetonitrile (B), both containing 0.1% formic acid. A binary gradient from 2% to 30% B was applied at a flow rate of 0.5 ml minute$^{-1}$, followed by a column wash and equilibration. The total run time was 18 minutes. All peptides were analyzed using electrospray ionization in positive mode (HESI source) using a full-scan data-dependent method with a range of m/z 200–2000. Other settings were: resolution (35,000), spray voltage (3.0 kV), capillary temperature (320˚C), heater temperature (350˚C), S-Lens RF Level (50 V), AGC target (1e6), and maximum IT (150 ms). The top 5 ions were subjected to data-dependent scans at a normalized collision energy of 15, 25 and 35. XCalibur software version 3 (ThermoScientific) was used for data acquisition. Data analysis was performed manually.

Sequences of 83 avian collagen 1α2 mRNA or cDNA entries were obtained from the NCBI nucleotide database (https://www.ncbi.nlm.nih.gov/nuccore/advanced), accessed in September 2021 (see Table 1). We preferred to use database nucleotide sequences, due to their higher reliability compared to protein sequences. For comparison, crocodile, turtle, snake,

**Table 1. Overview of collagen 1α2 data sources.** The species marked with an asterisk were included in the common bird ancestor estimation.

| Animal species | Common name | Bird order | Bird family | Data source |
|---|---|---|---|---|
| *Aquila chrysaetos chrysaetos* * | Golden eagle | Accipitriformes | Accipitridae | XM_030009145.1 |
| *Haliaeetus albicilla* | White-tailed eagle | Accipitriformes | Accipitridae | XM_009930894.1 |
| *Haliaeetus leucocephalus* | Bald eagle | Accipitriformes | Accipitridae | XM_010570018.1 |
| *Anas platyrhynchos* * | Mallard | Anseriformes | Anatidae | XM_038173912.1 |
| *Anser cygnoides domesticus* | Domestic goose | Anseriformes | Anatidae | XM_013187030.1 |
| *Aythya fuligula* | Tufted duck | Anseriformes | Anatidae | XM_032180715.1 |
| *Cygnus atratus* | Black swan | Anseriformes | Anatidae | XM_035545549.1 |
| *Cygnus olor* | Mute swan | Anseriformes | Anatidae | XM_040549604.1 |
| *Oxyura jamaicensis* | Ruddy duck | Anseriformes | Anatidae | XM_035317175.1 |
| *Chaetura pelagica* * | Chimney swift | Apodiformes | Apodidae | XM_009994908.1 |
| *Calypte anna* | Anna's hummingbird | Apodiformes | Trochilidae | XM_008494149.2 |
| *Apteryx australis mantelli* * | North Island brown kiwi | Apterygiformes | Apterygidae | XM_013941291.1 |
| *Apteryx rowi* | Okarito kiwi | Apterygiformes | Apterygidae | XM_026078914.1 |
| *Antrostomus carolinensis* * | Chuck-will's-widow | Caprimulgiformes | Caprimulgidae | XM_010177896.1 |
| *Cariama cristata* * | Red-legged seriema | Cariamiformes | Cariamidae | XM_009708994.1 |
| *Dromaius novaehollandiae* * | Emu | Casuariiformes | Casuariidae | XM_026098592.1 |
| *Charadrius vociferus* * | Killdeer | Charadriiformes | Charadriidae | XM_009895351.1 |
| *Calidris pugnax* | Ruff | Charadriiformes | Scolopacidae | XM_014940271.1 |
| *Columba livia* * | Rock dove | Columbiformes | Columbidae | XM_005504926.2 |
| *Merops nubicus* * | Northern carmine bee-eater | Coraciiformes | Meropidae | XM_008949969.1 |
| *Cuculus canorus* * | Common cuckoo | Cuculiformes | Cuculidae | XM_009558866.1 |
| *Eurypyga helias* * | Sunbittern | Eurypygiformes | Eurypygidae | XM_010160497.1 |
| *Falco cherrug* * | Saker falcon | Falconiformes | Falconidae | XM_005432123.3 |
| *Falco naumanni* | Lesser kestrel | Falconiformes | Falconidae | XM_040591801.1 |
| *Falco peregrinus* | Peregrine falcon | Falconiformes | Falconidae | XM_005228784.3 |
| *Falco rusticolus* | Gyrfalcon | Falconiformes | Falconidae | XM_037385294.1 |
| *Numida meleagris* * | Helmeted guineafowl | Galliformes | Numididae | XM_021387647.1 |
| *Centrocercus urophasianus* | Greater sage-grouse | Galliformes | Phasianidae | XM_042817165.1 |
| *Lagopus leucura* | White-tailed ptarmigan | Galliformes | Phasianidae | XM_042888556.1 |
| *Coturnix japonica* | Japanese quail | Galliformes | Phasianidae | XM_015853543.1 |
| *Gallus gallus* | Chicken | Galliformes | Phasianidae | NM_001079714.2 |
| *Phasianus colchicus* | Common pheasant | Galliformes | Phasianidae | XM_031614315.1 |
| *Gavia stellata* * | Red-throated loon | Gaviiformes | Gaviidae | XM_009808201.1 |
| *Balearica regulorum gibbericeps* * | Grey crowned crane | Gruiformes | Gruidae | XM_010300147.1 |
| *Leptosomus discolor* * | Cuckoo roller | Leptosomiformes | Leptosomidae | XM_009953335.1 |
| *Mesitornis unicolor* * | Brown mesite | Mesitornithiformes | Mesitornithidae | XM_010182829.1 |
| *Tauraco erythrolophus* * | Red-crested turaco | Musophagiformes | Musophagidae | XM_009985626.1 |
| *Opisthocomus hoazin* * | Hoatzin | Opisthocomiformes | Opisthocomidae | XM_009936134.1 |
| *Chlamydotis macqueenii* * | MacQueen's bustard | Otidiformes | Otididae | XM_010127319.1 |
| *Corvus brachyrhynchos* * | American crow | Passeriformes | Corvidae | XM_008629908.2 |
| *Corvus cornix cornix* | Hooded crow | Passeriformes | Corvidae | XM_039548651.1 |
| *Corvus kubaryi* | Mariana crow | Passeriformes | Corvidae | XM_042043192.1 |
| *Corvus moneduloides* | New Caledonian crow | Passeriformes | Corvidae | XM_032111992.1 |
| *Lonchura striata domestica* | White-rumped munia | Passeriformes | Estrildidae | XM_021538400.2 |
| *Taeniopygia guttata* | Zebra finch | Passeriformes | Estrildidae | XM_032746668.2 |
| *Serinus canaria* | Atlantic canary | Passeriformes | Fringillidae | XM_009085961.3 |
| *Hirundo rustica* | Barn swallow | Passeriformes | Hirundinidae | XM_040085392.1 |
| *Molothrus ater* | Brown-headed cowbird | Passeriformes | Icteridae | XM_036406384.1 |

*(Continued)*

**Table 1.** (Continued)

| Animal species | Common name | Bird order | Bird family | Data source |
|---|---|---|---|---|
| *Motacilla alba alba* | White wagtail | Passeriformes | Motacillidae | XM_038128293.1 |
| *Ficedula albicollis* | Collared flycatcher | Passeriformes | Muscicapidae | XM_005041092.2 |
| *Parus major* | Great tit | Passeriformes | Paridae | XM_015619636.2 |
| *Pseudopodoces humilis* | Ground tit | Passeriformes | Paridae | XM_005518806.1 |
| *Zonotrichia albicollis* | White-throated sparrow | Passeriformes | Passerellidae | XM_026796017.1 |
| *Onychostruthus taczanowskii* | White-rumped snowfinch | Passeriformes | Passeridae | XM_041408440.1 |
| *Pyrgilauda ruficollis* | Rufous-necked snowfinch | Passeriformes | Passeridae | XM_041466246.1 |
| *Chiroxiphia lanceolata* | Lance-tailed manakin | Passeriformes | Pipridae | XM_032694999.1 |
| *Corapipo altera* | White-ruffed manakin | Passeriformes | Pipridae | XM_027639216.1 |
| *Lepidothrix coronata* | Blue-crowned manakin | Passeriformes | Pipridae | XM_017817645.1 |
| *Manacus vitellinus* | Golden-collared manakin | Passeriformes | Pipridae | XM_008921186.3 |
| *Neopelma chrysocephalum* | Saffron-crested tyrant-manakin | Passeriformes | Pipridae | XM_027684680.1 |
| *Pipra filicauda* | Wire-tailed manakin | Passeriformes | Pipridae | XM_027749101.2 |
| *Sturnus vulgaris* | Common starling | Passeriformes | Sturnidae | XM_014875634.1 |
| *Camarhynchus parvulus* | Small tree finch | Passeriformes | Thraupidae | XM_030943434.1 |
| *Geospiza fortis* | Medium ground finch | Passeriformes | Thraupidae | XM_005418574.2 |
| *Catharus ustulatus* | Swainson's thrush | Passeriformes | Turdidae | XM_033071651.2 |
| *Empidonax traillii* | Willow flycatcher | Passeriformes | Tyrannidae | XM_027910641.1 |
| *Egretta garzetta* * | Little egret | Pelecaniformes | Ardeidae | XM_009641871.2 |
| *Pelecanus crispus* | Dalmatian pelican | Pelecaniformes | Pelecanidae | XM_009492035.1 |
| *Nipponia nippon* | Crested ibis | Pelecaniformes | Threskiornithidae | XM_009466601.1 |
| *Phaethon lepturus* * | White-tailed tropicbird | Phaethontiformes | Phaethontidae | XM_010282538.1 |
| *Picoides pubescens* * | Downy woodpecker | Piciformes | Picidae | XM_009907466.1 |
| *Fulmarus glacialis* * | Northern fulmar | Procellariiformes | Procellariidae | XM_009575128.1 |
| *Nestor notabilis* * | Kea | Psittaciformes | Nestoridae | XM_010017041.1 |
| *Melopsittacus undulatus* | Budgerigar | Psittaciformes | Psittaculidae | XM_034062147.1 |
| *Strigops habroptila* | Kakapo | Psittaciformes | Strigopidae | XM_030480918.1 |
| *Pterocles gutturalis* * | Yellow-throated sandgrouse | Pterocliformes | Pteroclidae | XM_010077375.1 |
| *Aptenodytes forsteri* * | Emperor penguin | Sphenisciformes | Spheniscidae | XM_009286338.1 |
| *Pygoscelis adeliae* | Adélie penguin | Sphenisciformes | Spheniscidae | XM_009330509.1 |
| *Athene cunicularia* * | Burrowing owl | Strigiformes | Strigidae | XM_026843207.1 |
| *Tyto alba* | Barn owl | Strigiformes | Tytonidae | XM_032991405.2 |
| *Struthio camelus australis* * | South African ostrich | Struthioniformes | Struthionidae | XM_009674272.1 |
| *Tinamus guttatus* * | White-throated tinamou | Tinamiformes | Tinamidae | XM_010212300.1 |
| *Apaloderma vittatum* * | Bar-tailed trogon | Trogoniformes | Trogonidae | XM_009867524.1 |
| *Common bird ancestor* | Common bird ancestor | Not applicable | Not applicable | constructed |
| *Crocodylus porosus* | Saltwater crocodile | Not applicable | Not applicable | XM_019533367.1 |
| *Pelodiscus sinensis* | Softshell turtle | Not applicable | Not applicable | XM_006114489.3 |
| *Python bivittatus* | Burmese python | Not applicable | Not applicable | XM_007425114.2 |
| *Pan troglodytes* | Chimpanzee | Not applicable | Not applicable | XM_001168894.5 |
| *Xenopus tropicalis* | Western clawed frog | Not applicable | Not applicable | NM_001079250.1 |
| *Oreochromis niloticus* | Nile tilapia | Not applicable | Not applicable | NM_001282897.1 |

mammalian, amphibian and fish collagen 1α2 sequences were added to the set, as well as a constructed common ancestral bird collagen 1α2 sequence. The sequences were translated to protein using Microsoft Excel version 2103. Collagen 1α2 sequences were only included in the data file if the GXY domain was 1014 codons in length (excluding the subsequent GGG triplet)

and if there was a glycine codon in each first GXY position, to promote the inclusion of high quality sequences [29]. The sequence of *Dromaius novaehollandiae* contained missing information, namely GGK at codon position 781 and CCY at position 1001. These codons were adapted to GGT and CCT, respectively, as these were the majority codons at the indicated positions. The bird species in the data set are from 33 orders. A coding DNA estimation of the collagen 1α2 GXY domain of the common bird ancestor was composed using the sequences of one species per order, marked with an asterisk in Table 1. Although there are differences in age between orders, the same weight was provided to each order to calculate the estimation. Of the 1014 positions, 1007 had majority codons (present in 17 or more of the 33 species) that were automatically selected for the ancestral sequence. There were 5 positions with a most abundant non-majority codon, that was selected. Finally, there were 2 positions that exhibited 2 non-majority codons of equal abundance. At position 611 (15x CCT, 15x CCC, 1x CCA and 2x GCC) CCC was selected for the ancestral sequence, because of its slightly higher probability compared to CCT when only single nucleotide changes are considered. At position 863 (16x AAC, 16x AGC and 1x AGT) AGC was selected for the same reason. The constructed common ancestral sequence of birds is reported in Fig 1 and was used to assess the genericity and suitability of bird collagen 1α2 targets.

## Results and discussion

### Theoretical target selection

The set of 83 bird collagen 1α2 sequences that met the selection criteria were visualized in codon, codon group and amino acid usage tables [29], to aid in the identification of a generic peptide (see S1 File). Together with several more distant species, the similarities of the birds' cDNA sequences are shown in Fig 2, by the number of mutual nucleotide differences. Additionally, it was calculated which amino acids were fully conserved, regarding the 83 species. After construction of the common bird ancestral cDNA sequence, see Fig 1, which was also included in the comparison of Fig 2, the ancestral sequence was translated to protein and *in silico* digested with trypsin, resulting in the formation of 79 peptides containing part of the GXY domain, as summarized in Table 2.

Analogue human collagen 1α2 (Uniprot entry P08123) contains 11 amino acids and 1 tryptic cleavage site N-terminal and 15 amino acids and 1 tryptic cleavage site C-terminal of the GXY domain. The selection of generic tryptic targets for birds was performed in two rounds. The first round focused mainly on genericity of the target, unambiguity (meaning the target is present in a single form) and analyzability; the second round was aimed at uniqueness versus non-bird species. The following criteria were applied during the first selection round:

A. The peptide target should contain a maximum of 1 N, Q or M residue. Whereas full target unambiguity is highly desirable for quantitative targets, for qualitative targets it may be allowed that the sequence contains an amino acid that can be partially modified, e.g. N or Q (deamidation) or M (oxidation). We chose to avoid peptide targets containing more than one amino acid that could be partially modified, due to the resulting increase in the number of possible forms to be monitored.

B. The length of the peptide should be at least 7 residues to provide sufficient uniqueness versus other species than birds.

C. The peptide target should contain a maximum of 1 amino acid that has not been fully conserved, with respect to the 83 bird species.

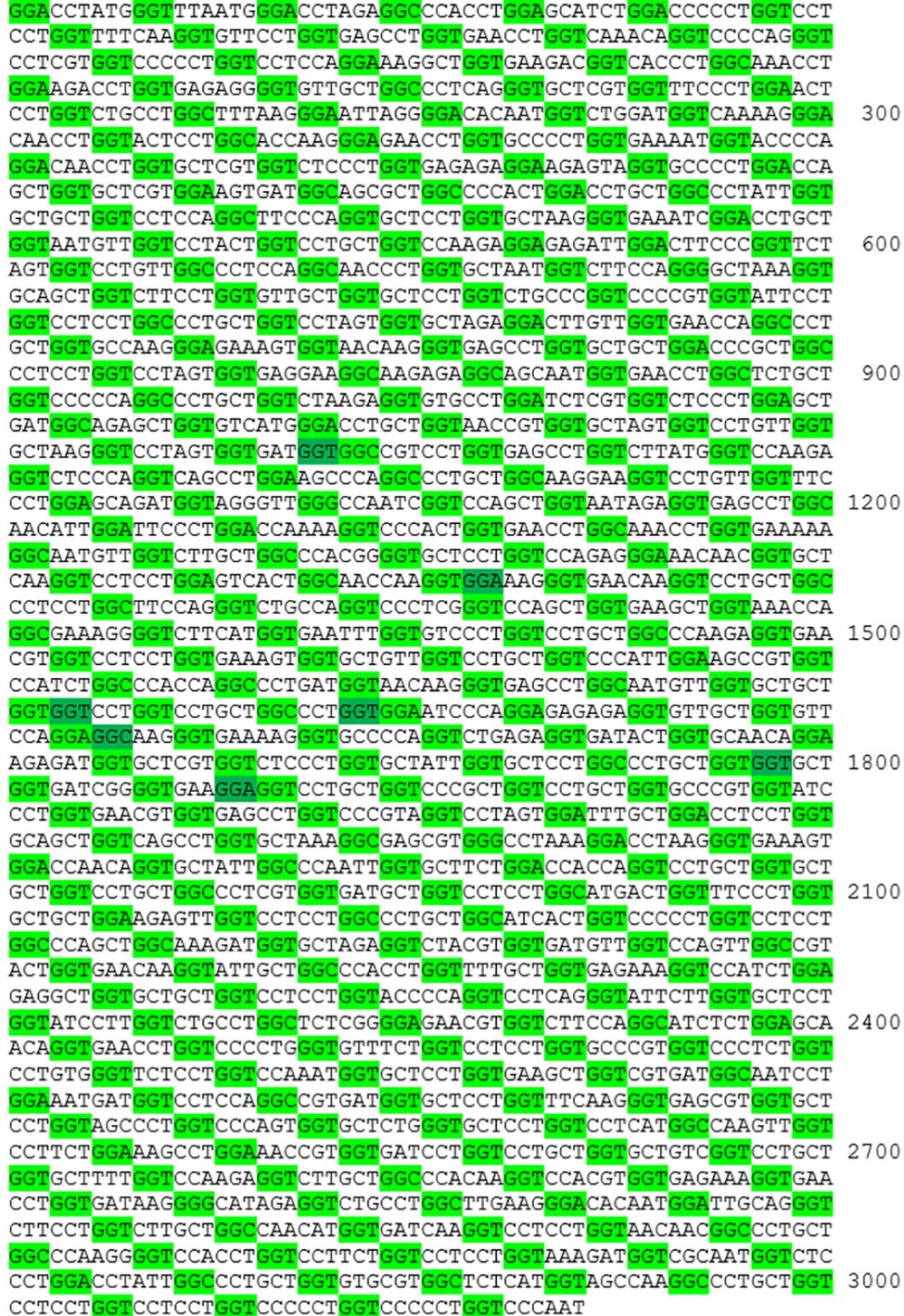

**Fig 1. Constructed coding DNA sequence of the common bird ancestor.** Constructed coding DNA sequence of the avian common ancestral collagen 1α2 GXY domain. Glycine codons in the 1st GXY position are highlighted in light green and other glycine codons in dark green.

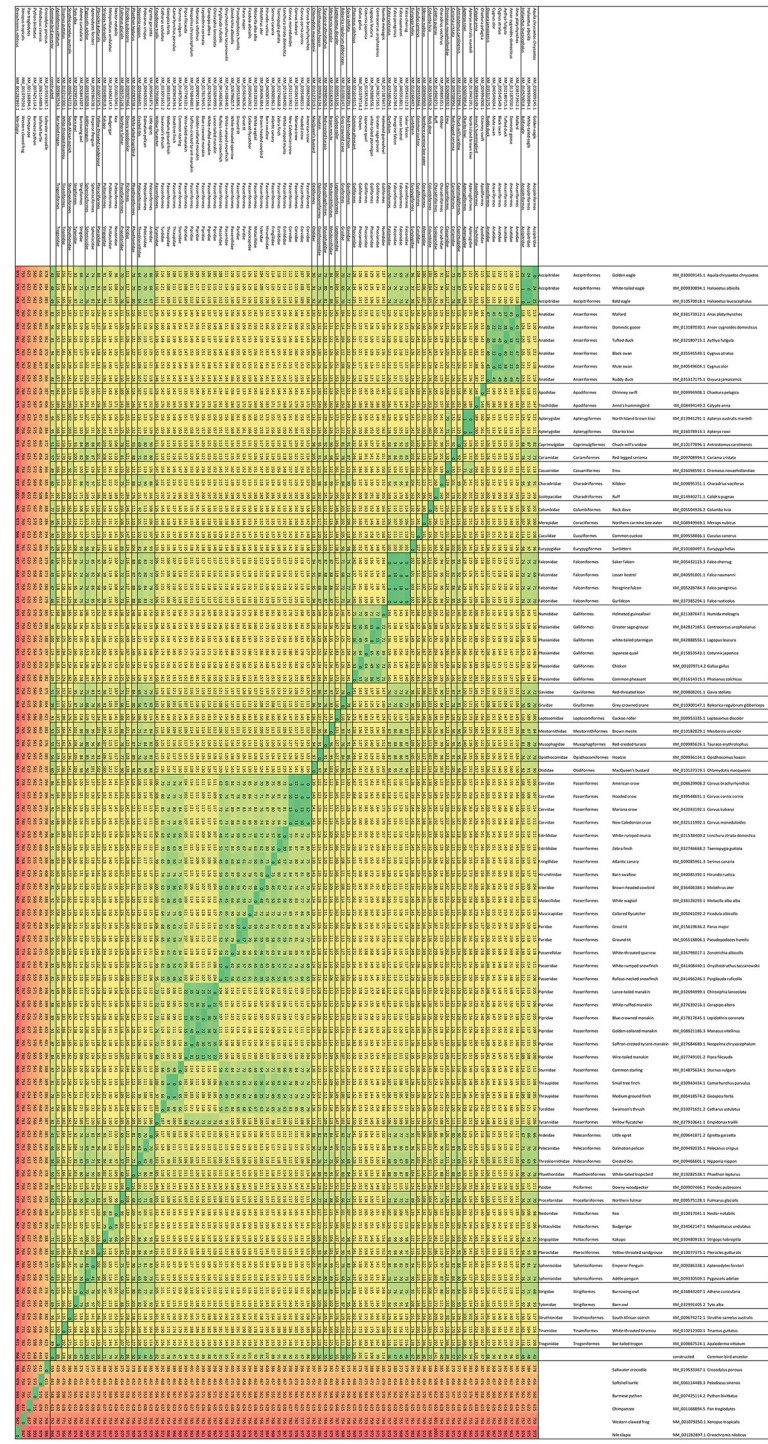

**Fig 2. Nucleotide differences between birds and other species.** Distance table of 90 compared animal species, including 83 bird species and the constructed common ancestor of birds, with respect to the collagen 1α2 GXY domain, indicating, from green via yellow to red, the increasing amount of nucleotide differences.

**Table 2. Results of the theoretical assessment of tryptic peptides.** The peptides were derived from the constructed common bird ancestor collagen 1α2 GXY domain. Assignments are explained in the main text.

| No | Sequence | Assignment | Remarks about tryptic peptide |
|---|---|---|---|
| 1 | GPMGLMGPR | A2 | N-terminal 5 or 6 additional AA |
| 2 | GPPGASGPPGPPGFQGVPGEPGEPGQTGPQGPR | C7 | |
| 3 | GPPGPPGK | D3 | |
| 4 | AGEDGHPGKPGRPGER | D2 candidate | Many hits with other animal species |
| 5 | GVAGPQGAR | C2 | |
| 6 | GFPGTPGLPGFK | E candidate | Many hits with other animal species |
| 7 | GIR | B3 | |
| 8 | GHNGLDGQK | C2 | |
| 9 | GQPGTPGTK | C2 | |
| 10 | GEPGAPGENGTPGQPGAR | A2 | |
| 11 | GLPGER | B6 | |
| 12 | GR | B2 | |
| 13 | VGAPGPAGAR | C2 | |
| 14 | GSDGSAGPTGPAGPIGAAGPPGFPGAPGAK | C4 | |
| 15 | GEIGPAGNVGPTGPAGPR | C3 | |
| 16 | GEIGLPGSSGPVGPPGNPGANGLPGAK | A2 | |
| 17 | GAAGLPGVAGAPGLPGPR | E candidate | Many hits with other animal species |
| 18 | GIPGPPGPAGPSGAR | C4 | |
| 19 | GLVGEPGPAGAK | C3 | |
| 20 | GESGNK | B6 | |
| 21 | GEPGAAGPAGPPGPSGEEGK | C4 | |
| 22 | R | B1 | |
| 23 | GSNGEPGSAGPPGPAGLR | C3 | |
| 24 | GVPGSR | B6 | |
| 25 | GLPGADGR | E candidate | Many hits with other animal species |
| 26 | AGVMGPAGNR | C2 | |
| 27 | GASGPVGAK | C5 | |
| 28 | GPSGDGGRPGEPGLMGPR | C2 | |
| 29 | GLPGQPGSPGPAGK | C2 | |
| 30 | EGPVGFPGADGR | E candidate | Hits with crocodile and turtle species |
| 31 | VGPIGPAGNR | D2 candidate | Except pheasant, contains VGPIGAAGNR |
| 32 | GEPGNIGFPGPK | E candidate | Many hits with other animal species |
| 33 | GPTGEPGKPGEK | C2 | |
| 34 | GNVGLAGPR | C2 | |
| 35 | GAPGPEGNNGAQGPPGVTGNQGGK | C3 | |
| 36 | GEQGPAGPPGFQGLPGPSGPAGEAGKPGER | C2 | |
| 37 | GLHGEFGVPGPAGPR | C2 | |
| 38 | GER | B3 | |
| 39 | GPPGESGAVGPAGPIGSR | C4 | |
| 40 | GPSGPPGPDGNK | C2 | |
| 41 | GEPGNVGAAGGPGPAGPGGIPGER | C5 | |
| 42 | GVAGVPGGK | C2 | |
| 43 | GEK | B3 | |
| 44 | GAPGLR | B6 | |
| 45 | GDTGATGR | D4 | |
| 46 | DGAR | B4 | |

*(Continued)*

**Table 2.** (Continued)

| No | Sequence | Assignment | Remarks about tryptic peptide |
|---|---|---|---|
| 47 | GLPGAIGAPGPAGGAGDR | C3 | |
| 48 | GEGGPAGPAGPAGAR | C6 | |
| 49 | GIPGER | B6 | |
| 50 | GEPGPVGPSGFAGPPGAAGQPGAK | C2 | |
| 51 | GER | B3 | |
| 52 | GPK | B3 | |
| 53 | GPK | B3 | |
| 54 | GESGPTGAIGPIGASGPPGPAGAAGPAGPR | C6 | |
| 55 | GDAGPPGMTGFPGAAGR | D3 | |
| 56 | VGPPGPAGITGPPGPPGPAGK | C2 | |
| 57 | DGAR | B4 | |
| 58 | GLR | B3 | |
| 59 | GDVGPVGR | C2 | |
| 60 | TGEQGIAGPPGFAGEK | C3 | |
| 61 | GPSGEAGAAGPPGTPGPQGILGAPGILGLPGSR | C5 | |
| 62 | GER | B3 | |
| 63 | GLPGISGATGEPGPLGVSGPPGAR | C6 | |
| 64 | GPSGPVGSPGPNGAPGEAGR | C5 | Contains partial cleavage site R20 |
| 65 | DGNPGNDGPPGR | | |
| 66 | DGAPGFK | E candidate | Many hits with other animal species |
| 67 | GER | C10 | Contains partial cleavage site R3 |
| 68 | GAPGSPGPSGALGAPGPHGQVGPSGKPGNR | | Contains partial P27 (after cleavage site K26) |
| 69 | GDPGPAGAVGPAGAFGPR | C3 | |
| 70 | GLAGPQGPR | D2 candidate | Many hits with other animal species |
| 71 | GEK | B3 | |
| 72 | GEPGDK | B6 | |
| 73 | GHR | B3 | |
| 74 | GLPGLK | B6 | |
| 75 | GHNGLQGLPGLAGQHGDQGPPGNNGPAGPR | C2 | |
| 76 | GPPGPSGPPGK | D2 candidate | Many hits with other animal species |
| 77 | DGR | B3 | |
| 78 | NGLPGPIGPAGVR | C2 | |
| 79 | GSHGSQGPAGPPGPPGPPGPPGPN | C5 | C-terminal 14 additional AA |

D. A maximum of 2 amino acid variations were allowed in the single non-conserved amino acid (see criterion C), again to limit the number of possible forms to be monitored and to reduce the probability that bird species that are not part of the set of 83, would show unexplored variations.

The results of the target selection are summarized in Table 2. For each of the 79 tryptic peptides of the constructed common bird ancestral sequence, the assignment is given in the third column. The assignments consist of a letter A to D, indicating which of the aforementioned criteria was not met (not exhaustive) and a number. For A assignments the number indicates the amount of N, Q or M amino acids in the sequence, for B assignments it indicates the peptide length, for C assignments the number of non-conserved residues in the peptide and for D assignments the number of amino acid variations in the single non-conserved residue. Fully conserved peptides were designated E. Only D2 and E assignments entered the second round

of selection. It should be noted that all D2 and E peptides also exhibited fully conserved preceding cleavage sites and absence of P in the amino acid C-terminal to the peptide, which is essential for their release from the primary structure by trypsin digestion. There were a total of 4 D2 and 6 E candidates. In the second round, the candidate peptides were subjected to protein blast search. The applied criterion was that no 100% hits should be obtained with animal species other than bird species. It was expected that hits with non-avian species would be obtained, especially for the E candidates. Hits with many other species were obtained for 5 out of 6 E candidates. However, the E candidate EGPVGF**p**GADGR had a limited number of hits, only with several species of crocodiles and turtles. It should be noted that **p** represents hydroxyproline which is often present at the third GXY position. For 3 out of 4 D2 candidates hits with many non-avian species were obtained, while there were no hits for the peptide VGPIGPAGNR. The only two remaining candidates represent positions 375–386 and 387–396 of the GXY domain. Fig 3 illustrates why this part of the sequence is suitable as generic bird target. Positions 375–396 of the bird sequence are more similar to the sequences of crocodiles and turtles than to the sequence of other reported species, as expected from the evolutionary relations. The peptide EGPVGF**p**GADGR is exactly the same between the constructed common ancestor of birds and the crocodile and turtle sequences shown, but differs from the next

| Species | Common bird ancestor | Alligator sinensis | Alligator mississippiensis | Crocodylus porosus | Gavialis gangeticus | Pelodiscus sinensis | Terrapene carolina triunguis | Chrysemys picta bellii | Python bivittatus | Protobothrops mucrosquamatus | Thamnophis sirtalis | Pogona vitticeps | Anolis carolinensis | Gekko japonicus | Xenopus tropicalis | Xenopus laevis | Latimeria chalumnae | Oreochromis niloticus |
|---|---|---|---|---|---|---|---|---|---|---|---|---|---|---|---|---|---|---|
| Accession | Constructed | XM_006025895.2 | XM_006258452.3 | XM_019533367.1 | XM_019508435.1 | XM_006114489.3 | XM_026649450.2 | XM_005308560.4 | XM_007425114.2 | XM_029283258.1 | XM_014057490.1 | XM_020785159.1 | XM_008112535.2 | XM_015420217.1 | NM_001079250.1 | XM_018266288.2 | XM_006011624.2 | NM_001282897.1 |
| Species type | Bird | Alligator | Alligator | Crocodile | Gharial | Turtle | Turtle | Turtle | Snake | Snake | Snake | Lizard | Lizard | Gecko | Frog | Frog | Coelacanth | Fish |
| Position | | | | | | | | | | | | | | | | | | |
| 374 | AAG K | AAG K | AAG K | AAG K | AAG K | AAG K | AAG K | AAG K | AAA K | AAA K | AAA K | AAA K | AAG K | AAG K | AAA K | AAG K | AAA K | AAG K |
| 375 | GAA E | GAA E | GAA E | GAA E | GAA E | GAA E | GAA E | GAA E | GAA E | GAA E | GAA E | GAA E | GAA E | GAA E | GAA E | GAA E | GAG E | GAG E |
| 376 | GGT G | GGT G | GGT G | GGT G | GGT G | GGT G | GGT G | GGT G | GGT G | GGT G | GGT G | GGT G | GGC G | GGT G | GGC G | GGT G | GGA G | GGA G |
| 377 | CCT P | CCT P | CCT P | CCC P | CCC P | CCC P | CCC P | CCC P | CCT P | CCT P | CAA Q | GCT A | TCT $S_1$ | CTT $L_1$ | CCT P | CTT $L_1$ | CCT P | CCT P |
| 378 | GTT V | GTT V | GTT V | GTT V | GTT V | GTT V | GTT V | GTT V | GCT A | TCT $S_1$ | ACT T | GTT V | GTT V | TCT $S_1$ | GCT A | GCT A | GCT A | GCT A |
| 379 | GGT G | GGT G | GGT G | GGT G | GGT G | GGT G | GGT G | GGT G | GGT G | GGT G | GGT G | GGT G | GGT G | GGT G | GGT G | GGT G | GGT G | GGT G |
| 380 | TTC F | TTC F | TTC F | TTC F | TTC F | TTC F | TTC F | TTC F | GCA A | TCT $S_1$ | CCA P | CTG $L_1$ | CTG $L_1$ | CTG $L_1$ | CCT P | CCT P | GCC A | CCT P |
| 381 | CCT P | CCT P | CCT P | CCT P | CCT P | CCT P | CCT P | CCT P | CCT P | GCT A | GCT A | CCT P | CCT P | TCT $S_1$ | CAA Q | CAA Q | TTT F | TCT $S_1$ |
| 382 | GGA G | GGT G | GGT G | GGT G | GGT G | GGT G | GGT G | GGT G | GGA G | GGA G | GGA G | GGT G | GGT G | GGT G | GGT G | GGT G | GGT G | GGA G |
| 383 | GCA A | GCA A | GCA A | GCA A | GCA A | GCT A | GCC A | GCC A | CCT P | CCT P | CCT P | GCT A | CCT P | CCT P | ATT I | ATT I | CTT $L_1$ | CAA Q |
| 384 | GAT D | GAT D | GAT D | GAT D | GAT D | GAT D | GAT D | GAT D | GAT D | GAT D | GAT D | GAT D | GAT D | GAT D | GAG E | GAG E | GAA E | GAT D |
| 385 | GGT G | GGC G | GGT G | GGT G | GGT G | GGT G | GGT G | GGT G | GGT G | GGT G | GGT G | GGT G | GGT G | GGC G | GGT G | GGT G | GGA G | GGC G |
| 386 | AGG $R_2$ | AGA $R_2$ | AGA $R_2$ | AGA $R_2$ | AGA $R_2$ | AGA $R_2$ | AGA $R_2$ | AGA $R_2$ | CGC $R_1$ | CGC $R_1$ | CGC $R_1$ | CGT $R_1$ | CGT $R_1$ | CGA $R_1$ | CGT $R_1$ | CGT $R_1$ | CGC $R_1$ | CGC $R_1$ |
| 387 | GTT V | GTT V | GTT V | GTT V | GTT V | GTT V | GTT V | GTT V | TCT $S_1$ | TCT $S_1$ | TCT $S_1$ | GCT A | CCT P | CCT P | TCT $S_1$ | AGT $S_2$ | AAT N | AGT $S_2$ |
| 388 | GGG G | GGC G | GGC G | GGC G | GGC G | GGC G | GGC G | GGC G | GGC G | GGC G | GGC G | GGC G | GGT G | GGT G | GGT G | GGC G | GGC G | GGA G |
| 389 | CCA P | CCA P | CCA P | CCA P | CCA P | CCA P | CCA P | CCA P | CCA P | CCA P | CCA P | CCC P | CCA P | CCA P | CCT P | CCT P | CCA P | CCT P |
| 390 | ATC I | ACT T | ACT T | ATT I | ATT I | ACT T | ACT T | ACT T | GTT V | ACT T | ACT T | ATT I | GTT V | GTT V | GCT A | GCT A | CCC P | CCA P |
| 391 | GGT G | GGT G | GGT G | GGT G | GGT G | GGT G | GGT G | GGT G | GGG G | GGC G | GGC G | GGT G | GGC G | GGT G | GGT G | GGT G | GGT G | GGC G |
| 392 | CCA P | CCA P | CCA P | CCA P | CCA P | CCT P | CCA P | CCA P | CCA P | CCA P | CCA P | CCA P | CCA P | CCA P | GCA A | CCA P | CCA P | CCA P |
| 393 | GCT A | GCT A | GCT A | GCT A | GCT A | GCT A | GCT A | GCT A | GCT A | GCT A | GCC A | GCT A | GCT A | GCT A | TCT $S_1$ | TCT $S_1$ | ACT T | AGT $S_2$ |
| 394 | GGT G | GGT G | GGT G | GGT G | GGT G | GGT G | GGT G | GGT G | GGT G | GGT G | GGT G | GGT G | GGT G | GGT G | GGT G | GGC G | GGA G | GGA G |
| 395 | AAT N | GCA A | GCA A | GCA A | GCA A | GCA A | GCA A | CCA P | CCA P | CCA P | CCA P | CCA P | CCA P | CCA P | CCA P | CCA P | CCA P | CCT P |
| 396 | AGA $R_2$ | AGA $R_2$ | AGA $R_2$ | AGA $R_2$ | AGA $R_2$ | AGA $R_2$ | AGA $R_2$ | AGA $R_2$ | CGA $R_1$ | CGA $R_1$ | CGA $R_1$ | AGA $R_2$ | AGA $R_2$ | AGA $R_2$ | AGA $R_2$ | AGA $R_2$ | AGA $R_2$ | AGA $R_2$ |
| 397 | GGT G | GGT G | GGT G | GGT G | GGT G | GGT G | GGC G | GGC G | GGT G | GGT G | GGT G | GGT G | GGT G | GGT G | GGT G | GGT G | GGC G | GGC G |

**Fig 3. Species comparison of positions 374–397 of the collagen 1α2 GXY domain.** The left sequence is from the constructed common ancestor of birds and the other sequences are from several reptile, frog and fish species, for comparison. On the left of each cell the codon is mentioned and on the right of each cell the codon group [21], which corresponds to amino acid (without the subscript). Compared to the common ancestor of birds, yellow cells correspond to amino acid differences and orange cells to codon group differences that do not lead to amino acid differences. Tryptic cleavage sites are presented as thick lines between cells.

closest species groups of snakes, lizards and geckos as well as from the amphibian and fish species. When it can be excluded that a sample contains crocodile or turtle material, by tracing the product's origin, the peptide EGPVGF**p**GADGR is suitable as generic bird peptide, especially as it is fully conserved considering the 83 bird species listed in Table 1. Ideally, a generic bird peptide should not occur in crocodile and turtle. As can be seen from Table 2, no other fully conserved E peptide is available with respect to the 83 investigated birds, but the VGPIGPAGNR peptide is the most suitable D2 candidate. First, the constructed common ancestral peptide of birds differs crucially from crocodiles and turtles in that there is an N residue at the 9th position, whereas an A residue occurs in the crocodile and turtle species. Second, there is an I residue at the 4th position, which is often T in crocodiles and turtles. As mentioned previously, VGPIGPAGNR is not fully conserved as it occurs in 82 of the 83 bird species investigated. Only in *Phasianus colchicus* (common pheasant) the P residue at the 6th position has changed to an A residue, resulting in the sequence VGPIGAAGNR. The combination of VGPIGPAGNR and VGPIGAAGNR can therefore be used to generically investigate the presence of birds, as it covers the whole group and differs from other animal groups.

## Experimental assessment

Extracts from ostrich, goose, duck, turkey, chicken, pheasant, guinea fowl, pigeon, partridge and quail were part of the experimental data set. In Fig 4 MS/MS spectra are presented of digested chicken and pheasant extract, showing both versions of the D2 candidate peptide. VGPIGAAGNR was determined experimentally in pheasant, while VGPIGPAGNR was observed in all other analyzed birds, see the result summary in Table 3. All relevant chromatograms and MS/MS spectra are reported in S2 File. A complicating aspect of the combination VGPIGPAGNR/VGPIGAAGNR is that it contains asparagine, which can be deamidated [30], especially in highly processed foods, which negatively affects the unambiguity. Therefore, it is necessary to also monitor the deamidated forms, which are 1 Da higher in mass, when examining the presence of bird material in processed food samples, bringing the total number of forms to be monitored to 4. An advantage of VGPIGPAGNR/VGPIGAAGNR is that it is shorter than EGPVGF**p**GADGR, theoretically making the combination more suitable to detect collagen if it were present in a more hydrolyzed form. It should be emphasized that the peptides were selected from 83 species classified into 33 orders. Since it is not yet possible to obtain a complete overview of sequences of birds and other species, it is advisable to monitor EGPVGF**p**GADGR besides VGPIGPAGNR/VGPIGAAGNR when the presence of birds is investigated. In addition, (functionally irrelevant) amino acid changes may have occurred in individuals within a species or be fixed in any bird species, exemplified by the pheasant change to VGPIGAAGNR, which will diminish the suitability of a generic target. On the other hand, (processed) food products often contain material from numerous indivuals, which increases the suitability of a generic target.

After having determined the presence of EGPVGF**p**GADGR and VGPIGPAGNR/VGPIGAAGNR in the extracts of 10 different bird species, we decided to investigate the presence of the same peptides in processed food products: homemade chicken soup, chicken broth tablets and a beef broth tablet as negative control. The results of these analyses are also summarized in Table 3. As expected, VGPIGPAGNR and EGPVGF**p**GADGR were detected in the chicken food products, but not in the bovine food product. In Fig 5 chromatograms are presented of a chicken and beef broth sample to illustrate this. The chicken samples also showed a deamidated form of VGPIGPAGNR. Deamidated VGPIGPAGNR ([M+2H]$^{2+}$ ions = > m/z 469.756) has the same nominal mass as the 2nd isotope of VGPIGPAGNR ([M+2H]$^{2+}$ ions = > m/z 469.766). However, the molecular species are separated by their LC retention time, their

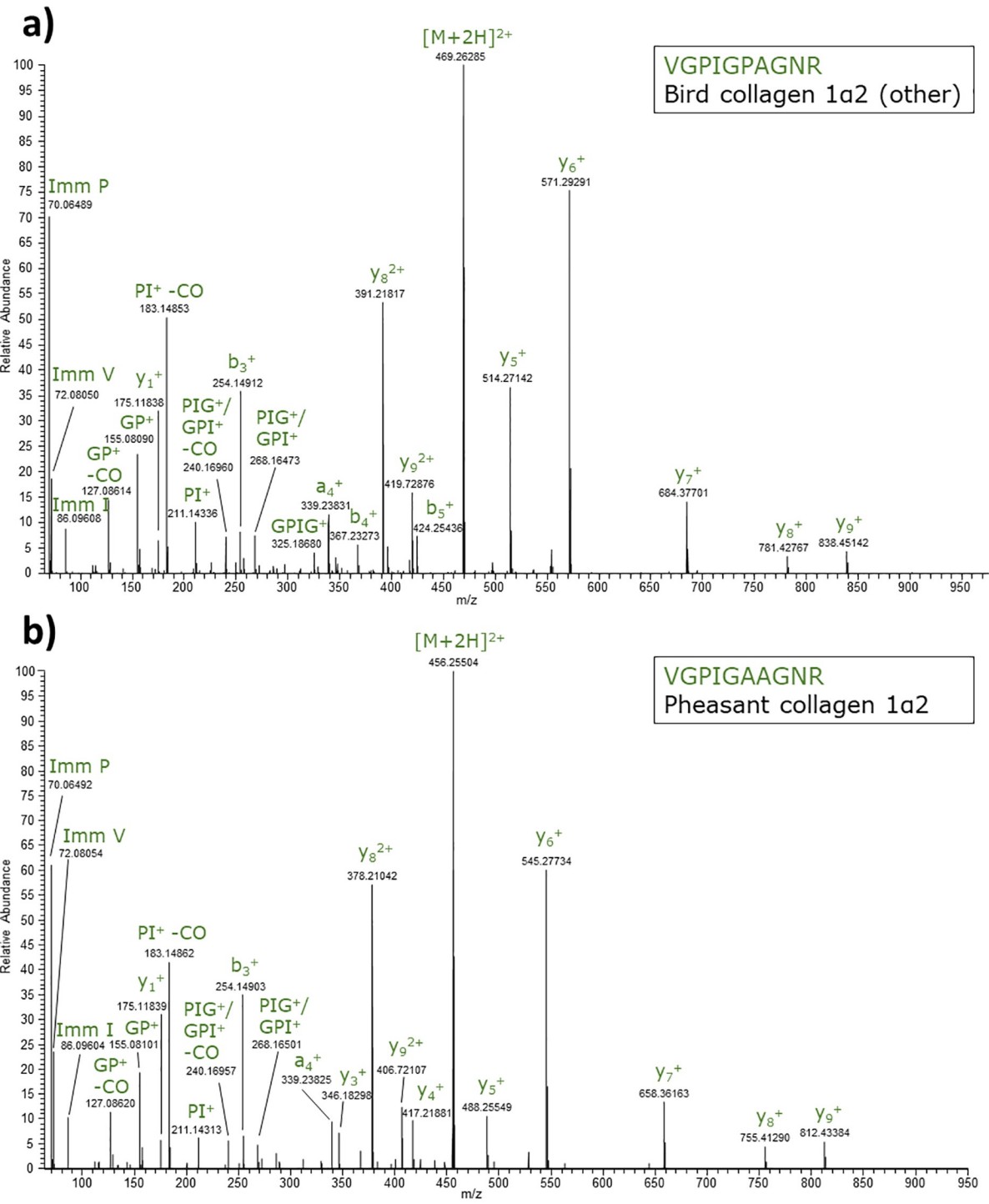

**Fig 4. MS/MS spectra of VGPIGPAGNR/VGPIGAAGNR.** MS/MS spectra of tryptic collagen 1α2 peptides from a) chicken (precursor m/z 469.26) and b) pheasant extract (precursor m/z 456.26). The chicken peptide VGPIGPAGNR is nearly generic for birds.

**Table 3. Results of the experimental assessment of generic bird peptides.** The presence of VGPIGPAGNR / VGPIGAAGNR (and/or deamidated) and EGPVGF**p**-GADGR in several bird samples and negative control beef broth is indicated.

| Sample | Detected peptide | | |
|---|---|---|---|
| | **VGPIGPAGNR (and/or deamidated)** | **VGPIGAAGNR (and/or deamidated)** | **EGPVGFpGADGR** |
| Ostrich tendon | V | - | V |
| Goose neck | V | - | V |
| Goose meat strip | V | - | V |
| Goose leg | V | - | V |
| Duck neck | V | - | V |
| Duck leg | V | - | V |
| Turkey neck | V | - | V |
| Turkey leg | V | - | V |
| Chicken leg | V | - | V |
| Pheasant meat strip | - | V | V |
| Pheasant leg | - | V | V |
| Guinea fowl torso | V | - | V |
| Pigeon torso | V | - | V |
| Partridge torso | V | - | V |
| Quail leg | V | - | V |
| Chicken soup | V | - | V |
| Chicken broth A | V | - | V |
| Chicken broth B | V | - | V |
| Beef broth | - | - | - |

exact mass and part of the fragment ions and thus can be easily distinguished. The bird peptides were clearly absent in beef broth. Instead, the bovine peptides GETGPAGPAGPIGPVGAR and GI**p**GEFGL**p**GPAGAR were detected, confirming the presence of bovine collagen 1α1 and 1α2 in the beef broth negative control sample. Finally, an MS/MS spectrum of EGPVGF**p**GADGR ([M+2H]$^{2+}$ ions = > m/z 587.778) obtained from chicken broth brand A is presented in Fig 6. The fragment ions were according to expectation: mainly b and y type ions, including water loss related to the N-terminal glutamic acid [31]. All relevant chromatograms and MS/MS spectra are reported in S2 File.

## Target coverage

Depending on the level of required genericity and uniqueness versus other animal species, it can be considered to use other peptides from Table 2 during an experiment. As an example the C2-assigned VGA**p**GPAGAR is discussed. The peptide contains 2 non-conserved amino acids, regarding the 83 investigated bird species, both of which exhibit two amino acid variations. The amino acid in the 1$^{st}$ position is V in 60 species and I in 23 species. The amino acid in the 3$^{rd}$ position is A in 82 species and G in 1 species, giving a total of 3 forms. The main form VGA**p**GPAGAR occurs in 60 species, including duck; the second most abundant form IGA**p**GPAGAR occurs in 22 species, including chicken. Finally only *Phaethon lepturus* or white-tailed tropicbird contains IGG**p**GPAGAR. The combination VGA**p**GPAGAR/ IGA**p**GPAGAR is sufficient to analyze chicken in duck and vice versa when it can be assumed that there are no other species present. When other (bird) species might be present, this combination is not suitable. Moreover, VGA**p**GPAGAR provides hits with many non-bird species (e.g. rat and mouse) and therefore the peptide is not suitable to generically investigate the presence of bird material in food products. In all cases, peptide targets should be

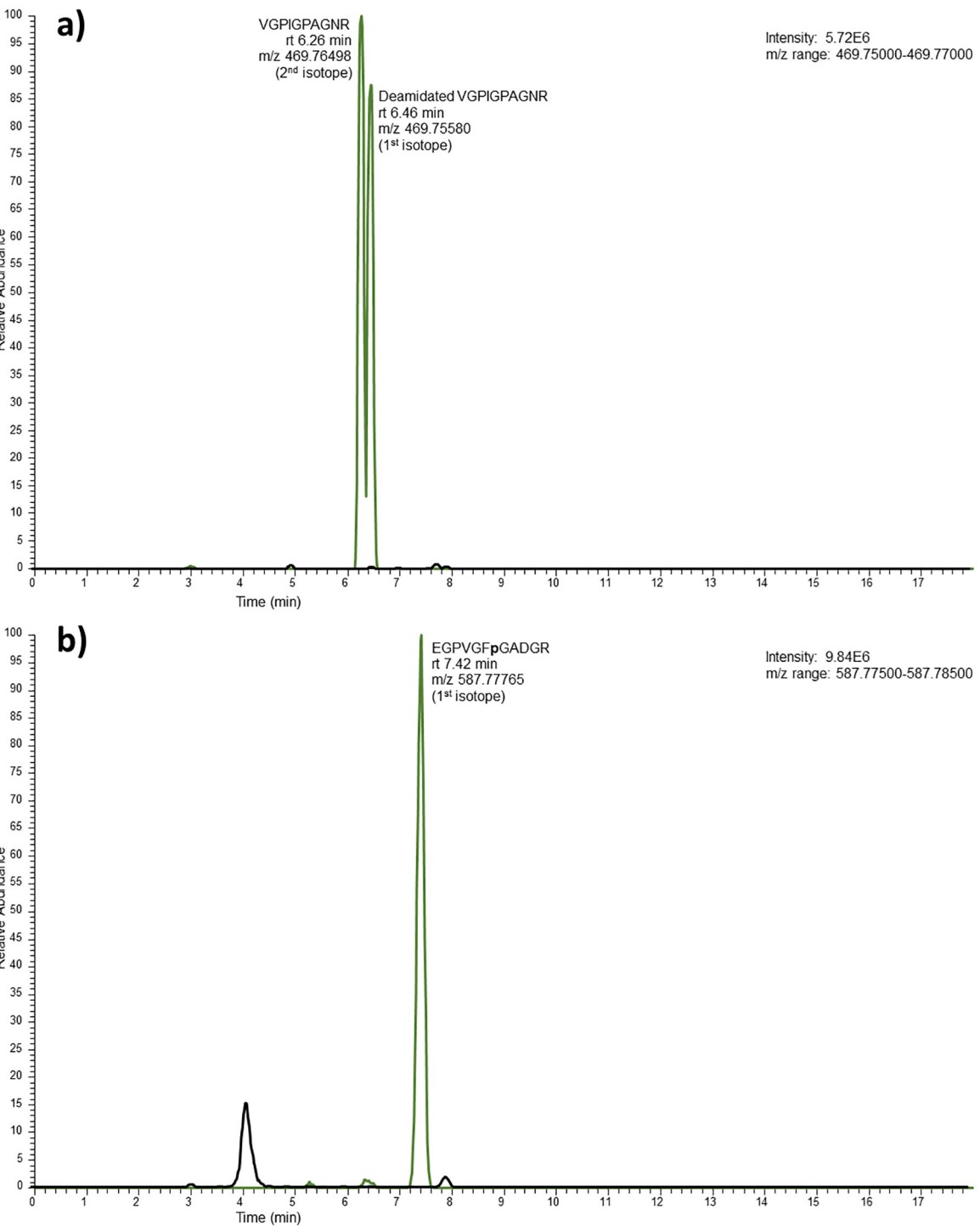

**Fig 5. Extracted chromatograms of bird targets.** Extracted chromatograms showing the signals for a) VGPIGPAGNR (2nd isotope) and deamidated VGPIGPAGNR (m/z 469.75 to 469.77) and b) EGPVGF**p**GADGR (m/z 587.775 to 587.785) in chicken (green lines, brand B) and beef (black lines) broth tablets.

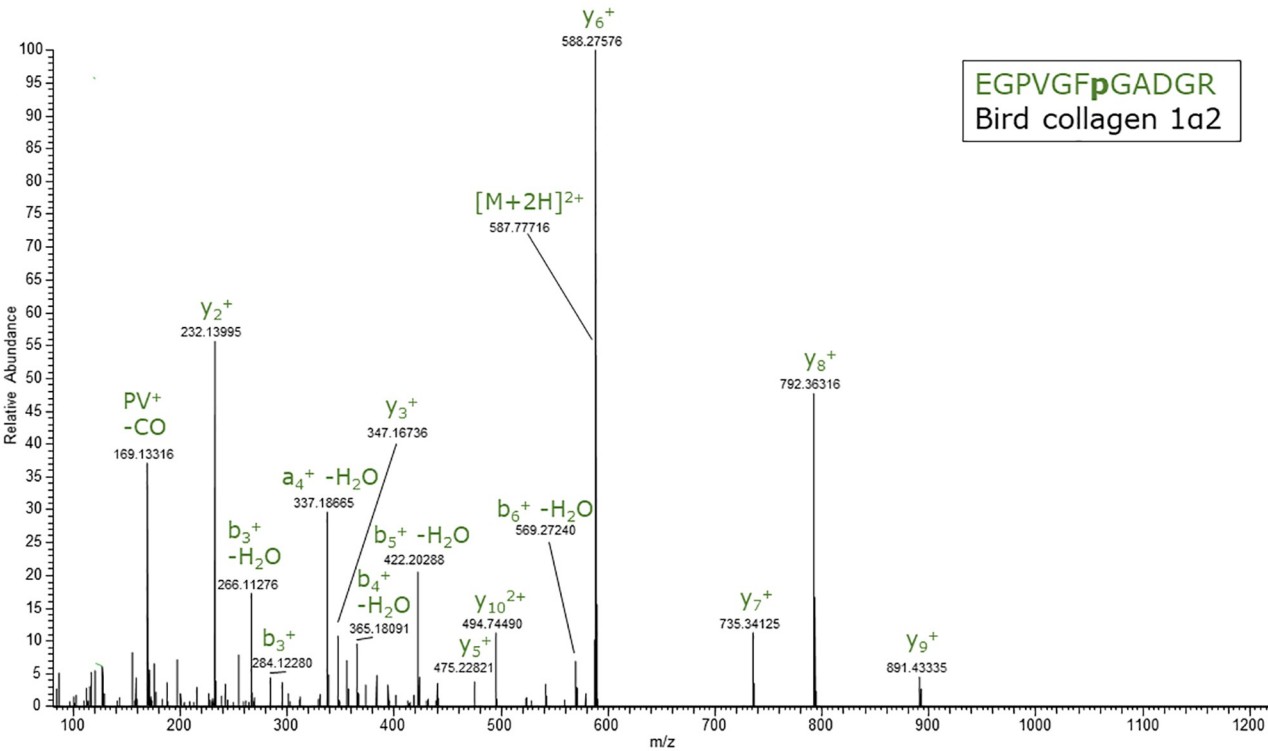

**Fig 6. MS/MS spectrum of EGPVGFpGADGR.** MS/MS spectrum of the tryptic collagen 1α2 peptide EGPVGF**p**GADGR (precursor m/z 587.78), which is generic for birds regarding the investigated bird species, but also occurs in crocodile and turtle species. The spectrum was obtained from chicken broth (brand A).

assessed regarding the required genericity and uniqueness, in relation to the goal of the study.

The peptide combination VGA**p**GPAGAR/IGA**p**GPAGAR is a clear example of a protein sequence part that could lead to confusing results when subjected to phylogenetic analysis. In a previous study it was established that changes in the GXY domain of type 1 collagens appear to be mainly the result of genetic drift and that back changes in a species' lineage can also occur within a functionally restricted space [29]. Whilst most of the bird orders investigated in the present study exhibited either the VGA**p**GPAGAR or IGA**p**GPAGAR form, the orders passeriformes, galliformes, apodiformes and anseriformes showed both forms within the species set. This could indicate that the two peptide forms have interchanged more than once during evolution and that the direction has not always been purely divergent. Alternatively, the peptide may not have been fully fixed to either form when the mentioned orders split off, which may persist even in the present, also for single species. Yet, fixation of a mutation is never an end result, exemplified by the white-tailed tropicbird sequence IGG**p**GPAGAR showing further divergence originating from a different position in the peptide.

## Collagen past, present and future

We presented several options to generically detect bird collagen 1α2, based on the current variation in the protein sequence. The divergence process, based mainly on genetic drift [29], will proceed in the future and, therefore, it is expected that more and more bird species will not contain the generic peptides presented in this study at some point in the future. Species will

inevitably drift further apart, although this process is very slow. For example, in humans (including the intermediate ancestors to a constructed common Euarchontoglires ancestor) an average of 1.1 nucleotide changes appear to have been fixed in the collagen 1α1 GXY domain (3042 nucleotides) per million years. This amounts to $3.5 \times 10^{-10}$ changes per nucleotide per year, excluding back changes [29]. Although the variation in collagen sequences within populations can be high [32], it is very rare for variations to become fixed. Moreover, only part of the nucleotide changes result in amino acid changes, which can be detected by protein LC-MS. The factor between nucleotide and amino acid changes is not constant, reducing the suitability of protein sequences for quantitative assessment of evolutionary relations [29]. Although collagen sequence changes in the more recent past appear to mainly have been governed by genetic drift, it is conceivable that future changes again will be more influenced by selective pressure, e.g. due to a change in external conditions. Such a change would have to be quite drastic to really affect the required collagen properties for tissue structure. Besides functional and informational restriction, the maintenance of code robustness [33] may also play a role during divergence, exemplified by codon usage bias structures. Overall, the current divergence status of bird collagen 1α2 makes it possible to apply generic detection using protein LC-MS. For fish species, however, it was observed that it does not seem feasible to select a comprehensive generic collagen type 1 target, due to the much longer divergence times [7]. The selection of generic peptides for fish species is still possible, but at a lower level, e.g. for fish families. Unfortunately, the taxonomy nomenclature of species types, such as birds or fish, is quite confusing because the organization levels "orders" and "families" do not represent similar divergence times for birds and fish. This effect is especially visible in protein domains that are mainly governed by genetic drift, but can be obscured in domains under high selective pressure.

To aid in finding candidate targets, a bird ancestral sequence was constructed, which estimates the collagen 1α2 GXY sequence of birds in the past, from which the present sequences have diverged. The constructed ancestral sequence may also be useful for identification of collagen in fossilized tissues of birds and related species using paleoprotein analysis [34] as it has been observed that collagens can be preserved for millions of years [11,12]. In a previous study [13], five collagen 1α2 peptide sequences were reported for a specimen of *Brachylophosaurus canadensis* (age 80 million years), a hadrosaur species, see Table 4.

Three of the five peptide sequences are exactly the same between *Brachylophosaurus canadensis* and the constructed common bird ancestor, and for one peptide there was a single threonine-alanine difference. The bottom peptide from Table 4 exhibited two differences, leucine-proline and proline-isoleucine. These types of changes are not unexpected in collagen 1α2. In a previous study, we found that, when amino acid changes occur, a selection of codon groups, such as A, P, V, T, $S_1$, and I, is predominantly involved in changes between closely related 1α2 collagens [35], as part of a larger change infrastructure. Leucines are slightly less involved. Again, hydroxyproline is often present at the third GXY position instead of proline. Therefore, we further investigated the reported MS/MS spectrum for *Brachylophosaurus canadensis*

**Table 4. Evaluation of reported *Brachylophosaurus canadensis* collagen 1α2 peptides.** The corresponding sequences from the constructed common ancestor of birds are in the right column. Amino acid differences are highlighted in yellow.

| B. canadensis 1α2 | Bird common ancestor 1α2 |
| --- | --- |
| GATGLPGVAGAPGLPGPR | GAAGLPGVAGAPGLPGPR |
| GSNGEPGSAGPPGPAGLR | GSNGEPGSAGPPGPAGLR |
| EGPVGFPGADGR | EGPVGFPGADGR |
| GEPGNIGFPGPK | GEPGNIGFPGPK |
| GLPGESGAVGPAGPPGSR | GPPGESGAVGPAGPIGSR |

GLPGESGAVGPAGP**p**GSR [12]. It was deduced that, depending on the obtained resolution, our constructed ancestral bird sequence GP**p**GESGAVGPAGPIGSR (which has exactly the same mass as the reported GLPGESGAVGPAGP**p**GSR), may fit the MS/MS data better than the original assignment, as it would explain the ions observed at nominal m/z 1424. These ions were assigned as "Potentially co-eluting contaminating ion" but could be assigned as $y_{16}^{+}$ ions of GP**p**GESGAVGPAGPIGSR. The unassigned peak at nominal m/z 712 could then represent $y_{16}^{2+}$ ions of GP**p**GESGAVGPAGPIGSR. This finding indicates that the construction of ancestral sequences using sequences of extant species could be helpful in the structural elucidation of paleoproteins, linking the research fields food chemistry, molecular evolution and paleontology, and providing a strong combination of disciplines to support paleontology in the 21st century and beyond.

## Conclusion

Generic LC-MS bird targets were identified after theoretical target selection, using a set of 83 bird collagen 1α2 sequences of 33 orders and a constructed common ancestral sequence, followed by experimental assessment of extracts from 10 bird species. Two tryptic target peptides passed the selection citeria, aimed at genericity, unambiguity, analyzability and uniqueness vs. non-bird species. The combination of VGPIGPAGNR and VGPIGAAGNR (pheasant only) covers all the investigated birds and was not found in other species using protein blast. It should be noted that it is necessary to monitor the deamidated forms in combination with the unmodified forms, as deamidation of N can occur during food processing. The peptide EGPVGF**p**GADGR covers all the investigated birds, but also occurs in several species of crocodiles and turtles. Only when the presence of the latter species can be excluded, the peptide is suitable as generic bird target. The presence of the generic peptide (combination) was confirmed in chicken soup and broth, with beef broth as negative control sample, providing proof of principle in food products. The constructed common ancestral bird sequence was also used to evaluate elucidated dino sequences, demonstrating that the use of ancestral sequences could be helpful in paleoprotein analysis.

## Supporting information

**S1 File. Calculations and data.** Calculations and data regarding the generic targets, the common bird ancestor and the distance table.
(XLSX)

**S2 File. Chromatograms and MS/MS spectra.** All relevant chromatograms and MS/MS spectra of the bird samples and negative control beef broth.
(PDF)

## Acknowledgments

Gerard Wolfis is acknowledged for preparing chicken soup.

## Author Contributions

**Conceptualization:** Anne J. Kleinnijenhuis, Frédérique L. van Holthoon.

**Data curation:** Anne J. Kleinnijenhuis.

**Formal analysis:** Anne J. Kleinnijenhuis, Frédérique L. van Holthoon.

**Funding acquisition:** Anne J. Kleinnijenhuis.

**Investigation:** Anne J. Kleinnijenhuis, Frédérique L. van Holthoon.

**Methodology:** Anne J. Kleinnijenhuis, Frédérique L. van Holthoon.

**Project administration:** Anne J. Kleinnijenhuis.

**Resources:** Anne J. Kleinnijenhuis, Frédérique L. van Holthoon.

**Software:** Anne J. Kleinnijenhuis, Frédérique L. van Holthoon.

**Supervision:** Anne J. Kleinnijenhuis.

**Validation:** Anne J. Kleinnijenhuis, Frédérique L. van Holthoon.

**Visualization:** Anne J. Kleinnijenhuis.

**Writing – original draft:** Anne J. Kleinnijenhuis, Frédérique L. van Holthoon.

**Writing – review & editing:** Anne J. Kleinnijenhuis, Frédérique L. van Holthoon.

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
