## [Decision Letter · Decision Letter 0]

4 Oct 2022

PONE-D-22-25317The quest for a generic bird target to detect the presence of bird in food products and considerations for paleoprotein analysisPLOS ONE

Dear Dr. Kleinnijenhuis,

Thank you for submitting your manuscript to PLOS ONE. After careful consideration, we feel that it has merit but does not fully meet PLOS ONE’s publication criteria as it currently stands. Therefore, we invite you to submit a revised version of the manuscript that addresses the points raised during the review process.

ACADEMIC EDITOR: Would you please go through the comments raised by the diligent reviewers and amend the MS accordingly. All corrections should be highlighted in red.. 

We look forward to receiving your revised manuscript.

Kind regards,

A. M. Abd El-Aty

Academic Editor

PLOS ONE

Journal Requirements:

"The authors received no specific funding for this work. The work was financed by Triskelion. No external funding was used." 

Reviewers' comments:

Reviewer's Responses to Questions

**Comments to the Author**

1. Is the manuscript technically sound, and do the data support the conclusions?

Reviewer #1: Yes

Reviewer #2: Partly

2. Has the statistical analysis been performed appropriately and rigorously? 

Reviewer #1: Yes

Reviewer #2: N/A

3. Have the authors made all data underlying the findings in their manuscript fully available?

Reviewer #1: Yes

Reviewer #2: No

4. Is the manuscript presented in an intelligible fashion and written in standard English?

Reviewer #1: Yes

Reviewer #2: No

5. Review Comments to the Author

Reviewer #1: The article needs some revision to be suitable for publication

-The abstract should be informative and include the main findings.

-Language should be revised.

-Figures should be represented in higher resolution

- The introduction should be enriched with recent references (2016-2022)

Reviewer #2: The manuscript “The quest for a generic bird target to detect the presence of bird in food products and considerations for paleoprotein analysis” by Kleinnijenhuis and van Holthoon is proposed as a typical untargeted approach for “protein biomarkers” identification.

However, with respect to the previous published papers (DOI: 10.1016/j.foodchem.2017.09.104 and 10.1016/j.fochx.2022.100333) some crucial steps are missing. In a classical untargeted approach, the “in silico” analysis is followed by an experimental validation of the candidate biomarkers. Here, the protein identification following tryptic digestion is missing. The authors suggest Xcalibur as software for data analysis even though it is only a software for raw data visualization. Some major points need to be addressed by the authors to improve the manuscript quality:

-The authors must perform a protein identification via Proteome Discover as in the already published papers by using a custom database. This step is crucial also to check the efficacy of the collagen extraction process;

-The authors clarify how they placed several grams per product in milliQ water for 2 days at 100°C and why they didn’t perform reduction/alkylation;

-MS/MS fragmentation spectra captions showing the precursor m/z are missing. Did you manually assign the sequence?

-A very careful reading of the manuscript and a review of the English language is recommended.

6. PLOS authors have the option to publish the peer review history of their article (what does this mean?). If published, this will include your full peer review and any attached files.

Reviewer #1: No

Reviewer #2: No

---

## [Author Response · Author response to Decision Letter 0]

14 Nov 2022

Our response has been uploaded in a separate file.

---

## [Decision Letter · Decision Letter 1]

7 Dec 2022

The quest for a generic bird target to detect the presence of bird in food products and considerations for paleoprotein analysis

PONE-D-22-25317R1

Dear Dr. Kleinnijenhuis,

We’re pleased to inform you that your manuscript has been judged scientifically suitable for publication and will be formally accepted for publication once it meets all outstanding technical requirements.

Kind regards,

A. M. Abd El-Aty

Academic Editor

PLOS ONE

Additional Editor Comments (optional):

Reviewers' comments:

Reviewer's Responses to Questions

**Comments to the Author**

1. If the authors have adequately addressed your comments raised in a previous round of review and you feel that this manuscript is now acceptable for publication, you may indicate that here to bypass the “Comments to the Author” section, enter your conflict of interest statement in the “Confidential to Editor” section, and submit your "Accept" recommendation.

Reviewer #1: (No Response)

Reviewer #2: All comments have been addressed

2. Is the manuscript technically sound, and do the data support the conclusions?

Reviewer #1: (No Response)

Reviewer #2: Yes

3. Has the statistical analysis been performed appropriately and rigorously? 

Reviewer #1: (No Response)

Reviewer #2: N/A

4. Have the authors made all data underlying the findings in their manuscript fully available?

Reviewer #1: (No Response)

Reviewer #2: Yes

5. Is the manuscript presented in an intelligible fashion and written in standard English?

Reviewer #1: (No Response)

Reviewer #2: Yes

6. Review Comments to the Author

Reviewer #1: The revised article (The quest for a generic bird target to detect the presence of bird in food products and considerations for paleoprotein analysis) is well written and could be accepted.

Reviewer #2: The Authors have addressed all of my concerns with the original manuscript. The revised manuscript is ready for publication

7. PLOS authors have the option to publish the peer review history of their article (what does this mean?). If published, this will include your full peer review and any attached files.

Reviewer #1: No

Reviewer #2: No

---

## [Editor Report · Acceptance letter]

12 Dec 2022

PONE-D-22-25317R1 

The quest for a generic bird target to detect the presence of bird in food products and considerations for paleoprotein analysis 

Dear Dr. Kleinnijenhuis:

I'm pleased to inform you that your manuscript has been deemed suitable for publication in PLOS ONE. Congratulations! Your manuscript is now with our production department. 

Kind regards, 

on behalf of

Prof. A. M. Abd El-Aty 

Academic Editor

PLOS ONE